# Super Secondary Structures of Proteins with Post-Translational Modifications in Colon Cancer

**DOI:** 10.3390/molecules25143144

**Published:** 2020-07-09

**Authors:** Dmitry Tikhonov, Liudmila Kulikova, Arthur Kopylov, Kristina Malsagova, Alexander Stepanov, Vladimir Rudnev, Anna Kaysheva

**Affiliations:** 1Institute of Mathematical Problems of Biology RAS-the Branch of Keldysh Institute of Applied Mathematics of Russian Academy of Sciences, 142290 Pushchino, Moscow Region, Russia; dmitry.tikhonov@gmail.com (D.T.); likulikova@mail.ru (L.K.); 2Institute of Theoretical and Experimental Biophysics, Russian Academy of Sciences, 142290 Pushchino, Moscow Region, Russia; volodyarv@mail.ru; 3V.N. Orekhovich Institute of Biomedical Chemistry, 119121 Moscow, Russia; a.t.kopylov@gmail.com (A.K.); kristina.malsagova86@gmail.com (K.M.); aleks.a.stepanov@gmail.com (A.S.)

**Keywords:** colorectal cancer, ultrahigh-resolution mass spectrometry, post-translational modifications, helical pair, protein structural motifs, supersecondary structure

## Abstract

New advances in protein post-translational modifications (PTMs) have revealed a complex layer of regulatory mechanisms through which PTMs control cell signaling and metabolic pathways, contributing to the diverse metabolic phenotypes found in cancer. Using conformational templates and the three-dimensional (3D) environment investigation of proteins in patients with colorectal cancer, it was demonstrated that most PTMs (phosphorylation, acetylation, and ubiquitination) are localized in the supersecondary structures (helical pairs). We showed that such helical pairs are represented on the outer surface of protein molecules and characterized by a largely accessible area for the surrounding solvent. Most promising and meaningful modifications were observed on the surface of vitamin D-binding protein (VDBP), complement C4-A (CO4A), X-ray repair cross-complementing protein 6 (XRCC6), Plasma protease C1 inhibitor (IC1), and albumin (ALBU), which are related to colorectal cancer developing. Based on the presented data, we propose the impact of the observed modifications in immune response, inflammatory reaction, regulation of cell migration, and promotion of tumor growth. Here, we suggest a computational approach in which high-throughput analysis for identification and characterization of PTM signature, associated with cancer metabolic reprograming, can be improved to prognostic value and bring a new strategy to the targeted therapy.

## 1. Introduction

Modern advances in systems biology and proteomics have increased interest in studying the role of post-translational modification (PTM) of proteins in pathology development mechanisms [1]. The role in the implementation of biological activity of the protein has been found to be due to the three-dimensional (3D) structure and folding features [2]. However, the modification of amino acid residues makes a significant contribution to the structural and functional diversity of proteins [3]. PTMs produce a significant effect on a variety of cellular processes (including, but not limited to, DNA reparation, signal transduction, energy generation, and consumption, etc.) through regulation of protein activities and determining supramolecular interactions. [4]. In this regard, attempts have been made to determine the role of PTM protein in the development of various pathologies [3]. Methods for detecting PTM can be divided into two groups, the first of which is unambiguous, identifies a specific type of PTM, and is aimed at detecting a wide range of PTMs (10 or more types) in proteins [5]. This group is represented by immune, radio, and fluorescence detection methods using affinity probes [6]. The second group is represented by mass spectrometric methods in combination with bioinformation algorithms for PTM prediction [7,8,9]. Therefore, the well-developed approach of “multidimensional identification of proteins” or “MudPIT”, which is HPLC in combination with mass spectrometry, is most effective for detecting several types of PTMs that are most common in nature [10].

Direct regulation of enzymes with diverse mechanisms is typically accomplished through the dynamic utilization of PTMs. Signal transducers responsible for the mobility of PTMs are guided the same way, thus, balancing and driving the comprehensive network of intracellular signal exchange in response to internal and external stimuli [11]. Generally, identification of a specific PTM or, more commonly, their profile in a large-scale proteomic study is purposed to associate them with pathophysiological mechanisms [12]. Dysfunction of phosphorylation makes the most pronounced input in cancer initiation and development since kinases and phosphatases inflect proteins function to provide the capability for the regulation of signals transduction [11].

The duet of lysine acetyltransferases and deacetylases delivers an acetyl group from acetyl-CoA to ε-*N*-lysine of proteins, thereby providing regulation of DNA replication and transcription through modification of histones as the most well-characterized process, but it may also impact on other proteins to perform their activities [11].

Regulation of protein uptake is generally accomplished through the attachment of the short ubiquitin moiety to ε-*N*-lysine residues. Recent reports demonstrated that malfunctioning of ubiquitination at any stage (transferring and ligation) may play a critical role in cells’ metabolic reprogramming and trigger cancer development [11].

Evidently, the regulation of cellular processes through diverse PTMs may produce a notable and comprehensive impact on the development of tumors. However, a further, but still unresolved, challenge is the integration of known data related to the role of PTMs for targeted managing and treatment of heterogeneous cancers. Apparently, only the advent of the profound integration of genomic data and large-scale characterization of PTMs functioning can yield a new tool for personalized diagnostics, bringing novel therapeutic strategies and approaches [11].

Among oncological pathologies, colorectal cancer (CRC) is one of the most dangerous non-infectious diseases in the world in terms of disability and mortality [13]. Therefore, studying PTM proteins associated with the development of colorectal oncopathology is of great importance. In the present study, a comparative mass spectrometric analysis of the protein composition was carried out considering PTMs of the plasma samples of healthy volunteers and samples of patients with CRC for three types of biologically significant modifications: phosphorylation, acetylation, and ubiquitination. A bioinformatics approach is proposed for the spatial description of the environment of identified amino acid PTMs in protein structures to determine the type of supersecondary motif containing the modification and to conduct conformational analysis. The spatial characterization of PTMs in proteins compared to a typical two-dimensional localization in the primary protein structure provides new data on the spatial location of the modified amino acid residue in the protein structure, its accessibility to the aqueous environment, and indicates a probable conformational change in the protein structure with PTM as a result of changing the biological activity of the protein.

In this study, we used the ultrahigh-resolution HPLC-MS/MS approach followed by bioinformatics analysis to identify and characterize PTM loci in protein structures associated with the development of colorectal cancer (CRC).

## 2. Results and Discussion

### 2.1. Identification of Helical Pairs in Protein Structures

Space-compact and stable motifs of the supersecondary structure of a protein consist of two α-helices connected by a constriction, the so-called helical pairs. Supersecondary structures include α-α-corners, α-α-hairpins, L-structures, and *V*-structures [14,15,16]. The structures of the helical pairs are compact in space and contain a hydrophobic core and polar shell. The side chains of residues recessed in the hydrophobic core are hydrophobic. Hydrophilic side chains or other polar groups of amino acid residues can be recessed into a hydrophobic core (or other hydrophobic environment) if they are involved in the formation of hydrogen or salt bonds [14,16]. Each α-helix has at least one hydrophobic residue per coil; in this case, hydrophobic residues are located on one side of the α-helix and form a continuous hydrophobic cluster [15].

To study the supersecondary structural motifs in protein molecules, we formulated rules for the recognition and selection of helical pairs (Appendix A). A point model is proposed that describes structures with four points in space (Figure 1). The structure of the helical pairs can be described by the coordinates of points *A*_1_, *A*_2,_
*B*_1_, and *B*_2_, which are the starting and ending points of the axes of two helixes ($\vec{A_{2}A_{1}}$ and $\vec{B_{1}B_{2}}$); the orientation of the helixes in space is characterized by the angle between these vectors cos ($\vec{A_{2}A_{1}}$).

Indeed, if both helixes are approximated by cylinders, on which helixes are formed by a thread passing through Cα atoms, then the beginnings and ends of the cylinder axes give four points that describe the helical pair. Figure 1B shows a geometric representation of a supersecondary structure formed by two helices and a constriction between them for a fragment of the albumin chain (Protein Data Bank (PDB) ID 1AO6, Cα: 443–481, Table 1). Each helix is presented in the form of a cylinder, and the axis of the cylinders is determined. The planes passing through the axis of the cylinders are also shown.

Important characteristics of helical pairs are helical distances, the torsion angle between the axes of the helixes, the number of amino acids between the helixes, the length of the helixes, and the area and perimeter of the helix projections. Analysis of the area and perimeter of the polygon of intersection of the projections of the helices of protein molecules is important in the study of structural motifs, since it directly indicates the strength of the interaction of the helices in the helical pairs. Therefore, helical pairs with nonzero values of the area and perimeter of the polygon of intersection of the projections of helixes are additionally stabilized due to internal interactions. Figure 1C shows the intersection of the two helices of the albumin fragment. It can be seen from the figure that the projections and axes of the helixes intersect. The values of the intersection area of the projections of the helixes and interplanar distance are indicated. Based on the calculated values and the angle between the axes of the helixes, it can be assumed that this helical pair is an α-α-corner (Appendix A).

In this study, for the first time, to the best of our knowledge, typical structural motifs of proteins containing PTMs were detected in blood samples of patients with CRC, and their conformational analysis was performed.

### 2.2. Recognition and Selection of Protein Structures

Using tandem mass spectrometric analysis (HPLC-MS/MS) of the protein composition of blood plasma samples from two comparison series (see Appendix A), patients with CRC (*n* = 28) and healthy volunteers (*n* = 41), proteins containing PTMs were analyzed. We identified a portion of 57 believable PTM peptides corresponding to 29 proteins. Further data curation left only 25 proteins containing 42 PTMs in total, and of them 20 proteins with 29 PTMs were finally recovered as the most related to the group of CRC patients [17]. Among them, a conformational analysis was performed on six proteins containing 12 peptides with modified amino acid residues, for which intact 3D structures were annotated in the Protein Database (see Appendix A).

A comparative conformational analysis of the diversity of the studied protein fragments between those unrelated by nature proteins allows us to determine the similarity of supersecondary motifs in which the studied sequences of peptides are localized, the “typicality” of the conformation of the secondary elements in the supersecondary structure, the stability of the supersecondary structure in an aqueous solution, and the availability of a local 3D portion of the protein, formed by the studied fragment on the solvent. For this, we solved the problem of determining a conformational template by forming samples of protein structures for each of the 12 identified peptides in which PTMs were detected. The samples of 3D structures for each tryptic peptide included 2–107 PDB protein structures (Table 1, N_prot_ column). The coordinates of the beginning and end of the supersecondary structure and the constituent elements of the secondary structure were determined. It was found for the first time, to our knowledge, that protein fragments containing sequences of target peptides in unrelated structures in the composition of the samples corresponded to the helical pair type [18,19]. In this case, the identified amino acid modifications were localized in one of the alpha helices of the helical pair or at the end of the alpha helix in close proximity to the irregular section (see Appendix A).

Table 1 summarizes the results of a generalized conformational analysis of samples of protein structures of the studied PTM peptides for subsequent characterization and determination of the type of helical pair. Namely, the protein structures selected in the PDB contained the target peptide (Nprot column), the number of occurrences of the peptide sequence in the selected proteins (Npass column), the types of secondary structures formed by this peptide in the proteins of each sample (SS of PTM via DSSP), the average value of the area of solvent-accessible surfaces for each sample of protein structures (AAS), and standard deviation (std).

Most of the identified modifications were localized in α-helices (H, G, I) or in constrictions close to α-helices (T, S, C) and exposed to the solvent (Table 1 column “SS of PTM via DSSP”). As mentioned above, the studied peptides were localized in helical pairs present in the solvent. The number of 3D protein structures differed between samples, and the vast majority of structures referred to α-proteins. For each target amino acid fragment, the number of selected protein molecules (Nprot column) was not equal to the number of occurrences of the indicated peptide in the selected proteins (Npass column). This is because the tryptic peptide can occur more than once (Npass ≥ Nprot) in protein sequences. The ACC and std columns for each protein sample contain information on the average value of solvent-accessible surface areas (ACC) and standard deviation (std). Nonzero values of the areas (more than 20 Å^2^) indicate that not all PTMs were in the hydrophobic core but were on the surface accessible to the solvent. Therefore, the PTM of amino acid residues in helical pairs in samples of protein structures were localized mainly in α-helices, or at the helix and constriction in a hydrophilic cluster and were not found in a hydrophobic cluster. PTM was found in protein structures as part of samples exposed to the aqueous environment (solvent).

PTM SEQQ is a fragment of the protein in which the PTM was identified; the locus of modification and type are indicated by Ac (K), P (S), P (Y), and glygly (K). Nprot is the number of protein molecules containing the studied fragment; the selected proteins are not differentiated into homologous and non-homologous. Npass is the number of occurrences of the fragment in the selected proteins. SS of PTM via DSSP-types of secondary structures formed by a fragment in the proteins found: *H*-α-helix; *E*-extended strand, participates in β ladder; *G*-helix 3_10_; T is hydrogen bonded turn; S is bend; C is a coil; ACC is the average surface area of a solvent identified by PTM accessible to the solvent, the number of water molecules in contact with this residue * 10 or residual water in Å^2^; std is the standard deviation.

### 2.3. Analysis of Structural Motifs of Protein Molecules Containing PTM Associated with the Development of Colorectal Cancer

The study of structural motifs containing PTM showed that all target peptides are components of a helical pair. For each structure, the amino acid sequence was determined, and inter-helical distances (minimum, r, and interplanar, d), angles between the axes of the helixes (torsion θ and two-dimensional ϕ), area (S), and perimeter of the intersection polygon of helix projections were calculated (see Table 2). Table 2 of structural motifs containing a PTM of six proteins (PDB AC: 1AO6, 1J78, 4ACQ, 5JPN, 1JEQ, and 5DU3) shows the amino acid sequences of the selected helical pairs. The lowercase letters indicate the amino acids that form the helix, the lowercase letters indicate irregular sections, and the identified type of modification is indicated in the specified place. Next, a separation was made in space of structural motifs. The secondary structure containing the modified amino acid is a helix located along the main chain immediately before it (first), and a secondary structure plus a helix after it (second). If the peptide is in an irregular section (constriction), one double-helical motif with two helixes adjacent to the chain and a constriction containing a modified amino acid between them is selected. For each motif, the constriction length (Np) between the helices is determined, and the coordinates of the atoms of occurrence of structural motifs containing the PTM (in chain A) and the coordinates of the identified amino acid (in brackets) are also indicated. By analyzing the calculated characteristics of helical pairs, it is possible to divide all selected pairs into subsets according to the type of helical pair. Therefore, a subset consisting of structural motifs of the α-α-corner type includes helical pairs with equal distances d and r, and their values are not particularly large, since α-α-angles are additionally stabilized due to the internal interaction of the helixes. In addition, these structures have nonzero (sufficiently large) values of the area, S, and perimeter of the polygon of intersection of the projections of the helixes. The banner can be any length. There are α-α-corners with short and long constrictions, but structures with short constrictions, 3–5 amino acids between coils, are the most studied and are more common. Helical pairs, depending on their parameters, belong to the corresponding structural motifs (see the column “Type of helical pair”). It was found that three structures in which PTMs are localized form a structural motif of the α-α-corner type, eight α-α-hairpins, three l-structures, and two *V*-structures. Among the structures studied, there are those that do not belong to any of the types known to us, but they form double-helical pairs.

We have previously shown that a structural motif of the α-α-corner type is an autonomously stable structure [20]. In this case, autonomous stability is understood as the stability of the spatial structure of the studied structural motif separately from the protein molecule in which this structure is found. A numerical experiment using the molecular dynamics method showed that the α-α-corner as an autonomous structure is stable in an aqueous medium.

For all studied helical pairs, including 12 peptides, analysis was conducted of the distribution of the area of the modified site available for the solvent (the number of water molecules in contact with this residue is * 10 or the residual water in Å2). Figure 2 shows histograms of the distribution of protein molecules containing various modifications, depending on the area of the surface available to the solvent before and after the modification. The possibility of studying the distribution appeared due to the selection of protein molecules containing the studied tryptic peptides from the helical pairs from the PDB bank. Such a search for each peptide corresponding to albumin gave us samples, each of which consisted of 106–107 protein molecules, corresponding to other proteins from three or six protein molecules. The histograms of the distribution of selected becks for ALBU are presented in Figure 1B for the remaining five identified molecules (Figure 1C). The blue lines correspond to the distribution of molecules before modification, and the red lines correspond to PTMs. As the histogram shows, the surface area of the modified amino acid residue accessible to the solvent always increased compared to that of the unmodified amino acid.

As shown in Figure 2, the studied helical pairs are characterized by a large accessible surface for the solvent at a level of 50 Å^2^ or more. Therefore, the average available area of the modified amino acid for similar helical pairs localized in the hydrophobic core of the alpha protein is 5–10 Å^2^. In this case, the accessible surface areas for helical pairs containing PTMs are increased by 10–100 Å^2^ compared to intact structures without modifications.

### 2.4. The Biological Significance of PTM in Target Proteins Detected in Samples of Patients with Colorectal Cancer

Tumors development is a multi-stage process. Signs of oncopathogenesis cover the regulation of proliferation, evasion of growth suppressors, suppression of apoptosis, replicative immortality, and activation of invasion and metastasis. Fundamentals of these processes are underlined in proteomic instability caused by the corrupted regulation of biological activity through a wide range of PTM options. Characterization of the spatial environment of the modified amino acid residues is obligatory for a deeper understanding of mechanisms emphasizing carcinogenesis in particular. Results of this issue may encourage the design of specific utility for validation and prediction of PTMs in proteins by comparative analysis of changes in the mass spectrometric features peptides.

In this study, we analyzed the spatial environment of the modified amino acids in proteins with the annotated 3D structures: vitamin d-binding protein (VDBP), alpha-2-macroglobulin (A2MG), complement C4-A (CO4A), X-ray repair cross-complementing protein 6 (XRCC6), plasma protease C1 inhibitor (IC1), and serum albumin (ALBU). Possible contributions in CRC development for the identified modifications are suggested in Table 3.

VDBP is involved in vitamin D transport and storage, scavenging the extracellular *G*-actin, and enhancement of the chemotactic activity of C5-alpha for neutrophils during inflammation. VDBP comprises three structurally similar domains. The first domain is targeted for binding with vitamin D metabolites (35–49 a.a.). The actin-binding site is located at 373–403 a.a., spanning parts of domains 2 and 3, and part of domain 1. The C5a/C5a des-Arg binding site is located at 130–149 a.a. Membrane binding sites are positioned at 150–172 a.a. and 379–402 a.a. [27].

We identified acetylation for two lysine residues (K370-ac and K114-ac) localized at different domains of VDBP in patients with stages I–II of CRC (Appendix A). Actin and membrane binding sites are directly adjacent to acetylated lysine residues of VDBP. Vitamin D metabolites and C5a/C5a des-Arg binding sites are at a considerable distance from the modified amino acids. Since structural changes to sites are meaningfully distal, we considered options for regulation of the functioning of all VTDB binding sites.

Epidemiological investigations suggested that there is a significant inverse association between circulating 25-hydroxyvitamin d (25(OH)d) and the risk of CRC developing [21,22]. However, little is known regarding the role of VDBP in colorectal carcinogenesis [21]. A dynamic actin cytoskeleton characterizes normal epithelial cells, and polymerization and depolymerization of actin filaments enable the cell shape to change during migration and mitosis. When becoming invasive, CRC cells lose regular actin organization and adhesion ability [23]. The invasive potential is fostered by C5a which facilitates proliferation and regeneration by attracting myeloid-derived suppressor cells and supporting tumor promotion [24]. Thus, it is reasonable to assume that acetylation of VTDB is a response reaction toward an increased level of 25(OH)d, actin, and C5a.

Lysine acetylation at K370-ac was detected for CO4A (Appendix A). This complement factor is responsible for effective amide-mediated binding with immune aggregates or protein antigens (position 1125 a.a.). The site of acetylated lysine and the site for binding with immune aggregates and antigens are located in different domains and spatially significantly spaced. Hence, we suggest that the observed acetylation is responsible for antigen-enhanced activity and emphasized by the increased escalation of the immune response throughout the tumor growth.

Protein XRCC6 is featured by serine phosphorylation at S77-P (Appendix A). It is believed that dimer of XRCC5/6 is probably involved in stabilizing broken DNA ends and bringing them together (position 31) [25]. Sites for DNA modification and binding are in the shared domain and in proximity to each other. Taking into account that cancer is triggered by a consequent series of genetic alterations leading to the corrupted mechanisms of proliferation, differentiation, death, and genomic stability, the role of mutations in and irregular modification of XRCC5/XRCC6 becomes highly suspicious in cancer developing [25]. It has been reported that when acetylated by p300 protein, XRCC5 promotes tumor growth through binding with a promoter region of COX2 (cyclooxygenase-2) turning its up-regulation (UniProtKB-P05155 (IC1_HUMAN) https://www.uniprot.org/uniprot/P05155). Therefore, it is suggested that phosphorylation may change XRCC5 binding affinity to the promoter region but with uncertain benefit.

For IC1, tyrosine phosphorylation was detected at site Y297-P (Appendix A). Activation of the C1 complex is under the control of the IC1 through the inactive stoichiometric complex with C1r or C1s elements. It plays a decisive role in regulation of complement system, fibrinolysis, and generation of kinins. IC1 is a very efficient inhibitor of FXIIa and inhibits chymotrypsin and kallikrein [26]. Although there is no strong data about the exact role of IC1 in tumor growth, it is highly probable that tyrosine phosphorylation may accompany oncogenesis through modulation activity of complement system elements.

Numerous modifications, including lysine acetylation, serine and tyrosine phosphorylation, andlysine ubiquitination, were recognized for ALBU (Appendix A). Apparently, ALBU is the most well characterized and scrutinized protein. It comprises three domains, seven fatty acid binding sites, and two major structurally selective small molecule sites. Site-I is often referred as the “warfarin site” and proposed for interactions with large, heterocyclic, and negatively charged hydrophobic ligands.

In contrast, site-II purposed for hydrogen-bonding and electrostatic interactions with typically small, aromatic ligands and carboxylic acids [26].

The diverse role of ALBU makes it an engaging predictor of different conditions. There are plenty of studies regarding the contribution of ALBU to cancer development, which are mainly focused on hypoalbuminemia. While facing cancer-induced stimuli, liver cells are inclined to overproduction of pro-inflammatory cytokines, including IL-6 and IL-1, that suppress production of ALBU [28,29,30]. Notwithstanding, hypoalbuminemia cannot be considered a prognostic factor in CRC because it does not have statistical significance when taken alone [31].

We established that in addition to serine and tyrosine phosphorylation, lysine acetylation was the most contributing PTM found in ALBU (Appendix A). Since acetyl-Coa is the main source for acetylation in cells, it should be noted that conversion to acetyl-CoA is depleted in cancer cells due to overexpression and delayed degradation of hypoxia-inducible factor 1-alpha (HIF-1α) [32], which upregulates pyruvate dehydrogenase (PDH) kinase-1 (PDK1) needed for inhibition of PDH [33]. Therefore, PDK1 inhibitors demonstrate satisfied efficiency in cancer treatment and management. Such medications restore the activity of PDH and, consequently, trigger apoptosis of tumor cells through p53 acetylation [34].

Unfortunately, there is no string evidence regarding the native origination of ALBU acetylation. However, numerous studies reported about non-enzymatic acetylation of ALBU which produces significant contribution in tumorigenesis. It was demonstrated that treatment of human serum with aspirin resulted in acetylation of ALBU on 26 lysine residues, including K199, K402, K519, and K545. It is supposed that such modifications are caused by the activity of serum butyrylcholinesterase [35]. Another study reported modification of up to 17 residues, including D1, K4, K12, Y411, K413, and K414. In this case, the experience was explained by the enhanced esterase activity of ALBU [36]. If Y411 was pre-treated with diisopropylfluorophosphate, the esterase activity of ALBU was irreversibly inhibited and the proteins did not undergo self-acetylation [36]. The effect of acetylation was significantly less pronounced in patients with high glucose levels, which was attributed to the impact of ALBU glycation, preventing esterase activity [37]. Therefore, it is suggested that the observed abundant modification of lysine residues in ALBU is caused by its increased esterase activity in patients with CRC.

Other types of detected modifications are questionable. It has been reported that phosphorylation of ALBU can occur non-specifically and, thus, create non-naturally originated identifications in proteomic assays [38]. On the other hand, enhanced non-specific phosphorylation of proteins, including ALBU in warfarin-binding site [39], is a known effect of the impaired cell signaling network, which significantly contributes in tumor growth and development [40]. Therefore, it seems that phosphorylation of ALBU does not play a role in cancer development due to its possible secondary origination, although the detection of this type of PTM in ALBU indicates a possible risk of cancer development.

## 3. Materials and Methods

### 3.1. Demography

The study was conducted on a cohort of 129 subjects, 41 of which were from the control group of healthy volunteers and 28 were patients with diagnosed CRC. Blood samples were provided by the N. N. Blokhin Russian Cancer Research Centre. Samples were collected into pre-chilled EDTA-2K tubes and centrifuged at 4 °C for 10 min at 1500 *g* after gentle agitation. Plasma fraction (1 mL) was transferred into the new clean cryovials (nominal volume is 2 mL) and frozen at −80 °C until the use [17].

All patients and healthy volunteers signed their written consent to participate in the study. This study was approved by the Local Research Ethics Committee of the Institute of Urology and Reproductive Health (Sechenov University) (Protocol No. 10–18 of 7 November 2018).

### 3.2. Sample Preparation for MS Analysis

A volume of 40 µL blood plasma was mixed with denaturation solution (0.1% deoxycholic acid sodium salt, 6% acetonitrile, and 75 mM triethylammonium bicarbonate; рН 8.5) to 500 µL finally.

Digestion was performed with trypsin (200 ng/µL supplemented in 30 mM acetic acid) in two stages: at the first stage trypsin was added at a ratio of 1:50 (*w/w*) and the reaction incubated for 3 h at 37 °C. At the second stage the enzyme was added at a ratio of 1:100 (*w/w*) and the reaction incubated at 37 °C for 12 h (approximately) [17]. Details of the sample preparation protocol are presented in the PRIDE Project PXD015163.

### 3.3. Mass Spectrometry Protein Registration

The analysis was conducted on high-resolution Q Exactive-HF mass spectrometer (Thermo Scientific, Waltham, MA, USA) with installed introduced nanospray ionization (NSI) ionization source (Thermo Scientific, USA). Selection of mass spectrometry parameters for data acquisition was the requirements of the Human Proteome Organization (HUPO Guidelines, bullet point 9, version 3.0.0, released 15 October 2019) [41] for minimal length of the detected peptide for consideration and justification of PE1 proteins (according to the Uniprot KB Classification).

Data acquisition was performed in a positive ionization mode in a range of 420–1250 *m/z* for precursor ions (with resolution *R* = 60 K) and in a range with the first recorded mass of 110 *m/z* for fragment ions (with resolution of *R* = 15 K). Precursor ions were accumulated for a maximum integration time of 15 ms and fragment ions were accumulated for a maximum integration time of 85 ms. Top 20 precursor ions with a charge state between *z* = 2 + to *z* = 4+ were collected in the ion trap and pushed to the collision cell for fragmentation in high-energy collision dissociation (HCD) mode. The activation energy was normalized at 27% for the *m/z* = 524, *z* = 2+ and ramped within ±20% from the installed value.

Analytical separation was performed on an Ultimate 3000 RSLC Nano UPLC system (Thermo Scientific, USA). Samples were quantitatively (2 μg) loaded onto the enrichment column Acclaim Pepmap^®^ (5 × 0.3 mm, 300 Å pore size, 5 µm particle size) and washed at a flow rate of 20 μL/min for 4 min by the loading solvent (2.5% acetonitrile, 0.1% formic acid, and 0.03% acetic acid). Following the loading stage, peptides were separated onto Acclaim Pepmap^®^ analytical column (75 µm × 150 mm, 1.8 µm particle size, 60 Å pore size) in a linear gradient of mobile phases A (water with 0.1% formic acid and 0.03% acetic acid) and B (acetonitrile with 0.1% formic acid and 0.03% acetic acid) at a flow rate of 0.3 μL/min. The following elution scheme was applied: the gradient started at 2.5% of B for 3 min and raised to 12% of B for the next 15 min, then to 37% of B for the next 27 min and to 50% for the next 3 min. The gradient was rapidly increased to 90% of B for 2 min and was maintained for 8 min at a flow rate of 0.45 μL/min. Enrichment and analytical columns were equilibrated in the initial gradient conditions for the next 13 min at a flow rate of 0.3 μL/min before the following sample run. Mass spectrometric measurements were performed using the the equipment of “Human Proteome” Core Facility (IBMC, Moscow, Russia). Obtained data were adapted for the search engine and uploaded to the ProteomeXchange Consortium via the PRIDE partner repository with the dataset identifier PXD015163.

### 3.4. Protein Identification and Criteria Selection for Post-translational Modifications

Adapted peak lists were identified using OMSSA (version 2.1.9, Proteomics Resource, Seattle, WA, USA) search engine [22]. Against a concatenated target/decoy proteins sequence database UniProtKB (88703 (target) sequences with the restricted taxonomy (Homo sapiens). The decoy sequences were populated by reversed-sequence algorithm the SearchGUI engine (release 3.1.16, Compomics, Gent-Zwijnaarde, Belgium).

Peptides were parsed with mass tolerance of 10.0 ppm for the MS1 (precursor) level and with a tolerance of 0.01 Da for the MS2 (fragment ions) level. Trypsin was set as a specific protease and a maximum of two missed (internal) cleavages were allowed. Modifications of acetyl (K), phospho (S), phospho (T), phospho (Y), and Gly-Gly (K) were selected as flexible. Peptides and proteins identifications were extracted from using PeptideShaker version 1.16.11 (Compomics) and validated at a 1.0% false discovery rate (FDR) estimated as a decoy hit distribution.

To eliminate probable false positive results due to concatenated search of several PTMs, we curated only those results that fit the following requirements: (a) at least 98% confidence for peptide identification, (b) at least 80% of peptide sequence coverage by fragmentation spectra, (c) at least 10 units of D-score for PTM probability.

### 3.5. Analysis of Post-translational Protein Modifications

In the present study, proteins containing tryptic peptides of a certain type of modification were selected from the PDB. The selected proteins belong to the class of alpha-helical and globular proteins. For each peptide, its own sample was created to determine the conformational template. Further, all motifs containing the studied tryptic peptides were selected from the database. For the recognition and selection of structural motifs, our previously developed method was used [20,21,23]. The secondary structure of proteins was determined by the Kabsch and Sander DSSP method [42]. Using the same program, the available contact surfaces with the solvent were determined. Definitions of important characteristics of structural motifs were described in previously published studies [18,19]. Visual analysis of the structures was carried out using the RasMol molecular graphics program [43].

## 4. Conclusions

Molecular function of proteins is determined by selectivity of their interactions with partner molecules. Such interactions require a fairly rigid spatial structure of the protein. Even small structural changes caused by PTMs may lead to a loss or switching in the protein’s biological activity. Thus, comprehension of the 3D environment of protein modification is necessary for understanding their functioning and contribution in pathophysiological processes.

We proposed a new approach to the study of identified amino acid PTMs in protein structures using a spatial depiction of their immediate environment in support with determination of structural motifs covering the identified modifications and analyzing their main features.

The approach allows to reevaluate the probabilistic structural changes caused by the detected PTMs thereby making predictions regarding the biological activity of the protein. In this study, conformational analysis and structural motifs with PTM moieties were determined in samples of patients with CRC. The structural analysis was based on the extraction of the identified peptides from the PDB source to use the extracted data as conformational templates for PTM characterization. The coordinates of the beginning and end of the sequences in the structures of the protein molecules were established. The types of structural motifs formed were determined. It was shown that, generally, tryptic peptides are part of helical pairs: α-α-corners, l-structures, and *V*-structures. For each structure, structural spatial dimensions and characteristics and the surfaces of the modified moiety available for water molecules were analyzed.

Using conformational templates, modified by PTM, a.a residues could be localized in stable supersecondary structures. It has been shown that helical pairs (α-α-corners, α-α-hairpins, *V*- and l-structures) are 22–66 a.a. in their size, and the modified amino acids were exposed to the aqueous environment and localized in one of the alpha helices or at the end of the alpha helix in close proximity to the irregular region in the hydrophilic cluster and were not found in the hydrophobic cluster.

We showed that the modified surface area, available for the solvent, is moderately superior to that for unmodified structures, which indicates a local conformational change, or the “respiration of the molecule.” Such changes are probable and perhaps do not lead to a straightened protein globule.

We suggest that the studied modifications are likely participate in the regulation of protein functional activity, since they are typically located near the sites accommodated for binding with a partner molecule. Therefore, conformational analysis of PTM-induced changes permits “imaging” of a protein surface and evaluation of the ability to modulate biological function of protein. Further, we intend to conduct a numerical experiment of molecular dynamics to establish the stability of structural motifs without and with modification.

In summary, we assume that discovering the exact role of PTMs in the sophisticated determination of biological processes is still challenging. We need to get cleverer integration of the known data about the implication of PTMs in cancer development and metabolism to offer new strategies of targeted therapy that are sufficient, but not so expensive. If the basis is correct, even low doses of the targeted medication can be efficient enough, and cancer can be eminently treatable.

## Figures and Tables

**Figure 1 molecules-25-03144-f001:**
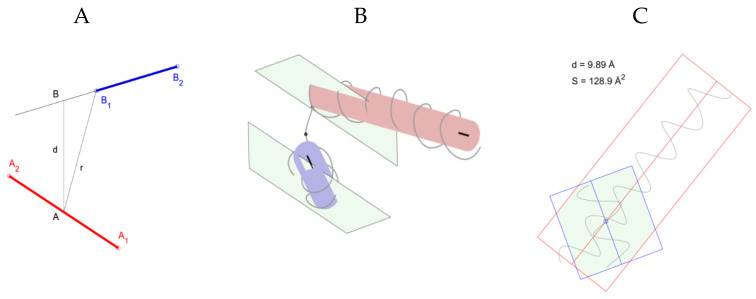
(**A**) Point model of a helical pair. The axis of the helical pair is shown. The segment (A1, A2) is the axis of the cylinder of the first helix, (B1, B2) is the axis of the cylinder of the second helix, d is the interplanar distance, and r is the minimum distance between the axes of the helixes. Helical pair for a 39 amino-acid-long albumin chain fragment (Protein Date Bank (PDB) ID 1AO6, plot coordinates: 443–481). (**B**) Approximate helix cylinders and planes passing through the axis of the cylinders. The curve is approximated by the positions of the Cα atoms of the protein chain, the atoms on the curve are indicated by dots, and (**C**) the intersection of the projections of the cylinders of the helixes of a helical pair. Polygon of intersection of helix projections for helical pair. The color indicates the area (S) of the polygon. The point of projection of the intersection of the axes of the helixes, the value of the interplanar distance (d).

**Figure 2 molecules-25-03144-f002:**
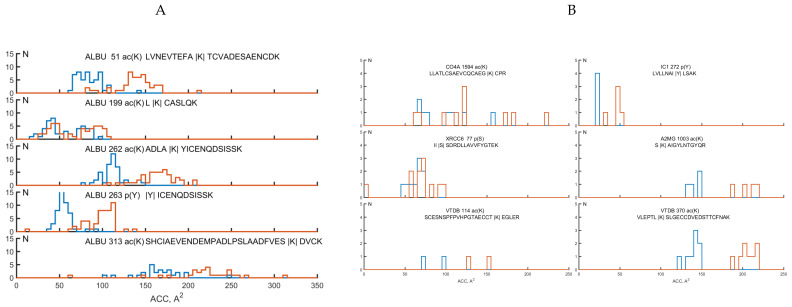
Histograms of the distribution of protein molecules for samples of protein structures containing various modification peptides, depending on the area of the accessible surface of the identified residue to the solvent before modification and with PTM. Each histogram is presented above: the name of the selected protein molecule (UniProt AC); the coordinates of the identified amino acid in this protein; type of modification (ac (K), p (S), p (Y)); and modified amino acid helical pair. The x-axis is the surface area of the solvent (unit of measurement is Å^2^). y-axis is the number of protein molecules containing the specified fragment. The blue lines correspond to the distribution of molecules before modification, and the red lines correspond to PTM. (**A**) ALBU: albumin, (**B**) complement C4-A: CO4A, IC1: plasma protease C1 inhibitor, XRCC6: X-ray repair cross-complementing protein 6, A2MG: alpha-2-macroglobulin, VTDB: vitamin d-binding protein.

**Table 1 molecules-25-03144-t001:** Structural parameters for proteins in samples of protein structures containing target amino acid fragments.

Protein Name	PTM SEQQ	Nprot	Npass	SS of PTM via DSSP	ACC, Å^2^	std
VTDB	vleptl|Ac(k)||slgeccdvedsttcfnak	6	8	H:8	136.8	8.8
VTDB	scesnspfpvhpgtaecct|Ac(k)||egler	6	2	S:2	82.0	17.0
CO4A	llatlcsaevcqcaeg|Ac(k)||cpr	6	10	C:4;G:2;S:3;T:1	91.3	30.5
XRCC6	ii|P(S)|sdrdllavvfygtek	6	11	T:11	57.0	20.8
IC1	lvllnai|P(Y)|lsak	3	5	E:5	23.8	4.7
A2MG	s|Ac(k)|aigylntgyqr	1	4	H:4	139.8	6.1
ALBU	l|Ac(k)|caslqk	107	157	H:157	48.0	20.4
ALBU	shciaevendempadlpslaadfves|Ac(k)|dvck	106	154	S:90;T:64	123.3	47.8
ALBU	adla|Ac(k)|yicenqdsissk	106	156	H:155;T:1	99.8	27.0
ALBU	|P(Y)|icenqdsissk	106	156	H:155;T:1	53.9	10.9
ALBU	lvnevtefa|Ac(k)|tcvadesaencdk	106	156	G:1;H:148;T:7	82.2	17.5

Abbreviations: PTM: post-translational modification; PTM SEQQ: fragment of the protein in which PTM was identified; Nprot: the number of protein molecules containing the studied fragment; Npass: the number of occurrences of the fragment in the selected proteins; SS of PTM via DSSP: types of secondary structures formed by this peptide in the proteins of each sample; ACC: solvent-accessible surface areas; std: standard deviation.

**Table 2 molecules-25-03144-t002:** Main characteristics of helical pairs containing post-translationally modified amino acids.

Protein Name	PDB AC	PTM SEQQ	Sequence of the Helical Pair	Locus	d	r	ϕ	θ	S	P	Np	Motif
VTDB	1J78	VLEPTL|Ac(k)|SLGECCDVEDSTTCFNAK	NTkvmdkytfelsRRTHLPevflskvleptl|k|slgEC	323–359 (354)	11.4	11.4	36.3	−35.9	148.1	48.1	6	α-α-corner
			LPevflskvleptl|k|slgECCDVEDsttcfnakgpllkkelssfidkgqelCA	340–392 (354)	9.0	9.3	13.5	−20.3	159.9	59.9	7	α-α-hairpin
		SCESNSPFPVHPGTAECCT|Ac(k)|EGLER	PGtaecCT|K|EglerklcmaaLK	106–127 (114)	0.7	8.7	13.5	−1.9	11.3	14.2	4	α-α-hairpin
			|K|EglerklcmaaLKHQPQEFPTYVEPTndeiceafrkDp	114–152 (114)	16.4	37.5	33.1	21.7	0	0	15	
CO4A	5JPN	LLATLCSAEVCQCAEG|Ac(K)|CPR	CSaevcqcaEG|K|CPRQRRALERGLQDEDGyrmkfacYY	1583–1620 (1594)	10.1	23.5	112.4	148.4	0	0	20	helical pair
XRCC6	1JEQ	II|P(s)|SDRDLLAVVFYGTEK	SKamfESQSEDELTpfdmsiqciqsvyiskii|S|S	45–78 (77)	0.49	4.42	109.8	22.91	1.64	6.93	9	helical pair
			LTpfdmsiqciqsvyiskii|S|SDRDLLAVVFYGTEKDKNSVNFKNIYVLQELDNPGakrileldQF	57–122 (77)	8.95	9.69	12.69	11.23	73.06	35.18	36	α-α-hairpin
IC1	5DU3	LVLLNAI|P(y)|LSAK	NNsdanlelintwvaknTNNKISRLLDSLPSDTRLVLLNAI|Y|LSAKWKTTFDpkkTR	231–287 (272)	8.0	30.9	79.7	−25.0	0	0	35	helical pair
A2MG	4ACQ	S|Ac(k)|AIGYLNTGYQR	XNmvlfapniyvldylneTQQLTpeiks|k|aigylntgyqrqln	975–1017(1003)	9.3	9.3	32.2	22.7	179.8	58.9	5	α-α-hairpin
ALBU	1AO6	L|Aс(к)|CASLQK	FYapellffakrykaaftecCQAADkaacllpkldelrdegkassakqrl|k|caslqkfGe	149–208 (199)	7.8	9.8	1.8	1.6	139.1	61.9	5	α-α-hairpin
			ADkaacllpkldelrdegkassakqrl|k|caslqkfGerafkawavarlsqrFP	172–224 (199)	2.3	5.0	52.1	−29.1	25.2	22.1	1	*V*-structure
		SHCIAEVENDEMPADLPSLAADFVES|Ac(k)|DVCK	SLaadfVES|K|DvcknyaeAk	304–323 (313)	0.6	8.1	105.4	38.9	0	0	5	l-structure
			|K|DvcknyaeAkdvflgmflyeyarRH	313–338 (313)	0.5	4.8	64.8	−7.3	15.2	16.1	1	*V*-structure
		ADLA|Ac(k)|YICENQDSISSK	AEfaevsklvtdltkvhteccHGDllecaddradla|k|yiceNq	226–268 (262)	6.1	7.2	19.7	16.7	90.5	53.7	3	α-α-hairpin
			GDllecaddradla|k|yiceNqdsiSS	248–273 (262)	1.2	4.6	99.0	33.4	5.2	9.3	1	l-structure
		|P(y)|ICENQDSISSK	AEfaevsklvtdltkvhteccHGDllecaddradlak|y|iceNq	226–268 (263)	6.1	7.2	19.7	16.7	90.5	53.7	3	α-α-hairpin
			GDllecaddradlak|y|iceNqdsiSS	248–273 (263)	1.2	4.6	98.9	33.4	5.2	9.3	1	l-structure
		LVNEVTEFA|Ac(k)|TCVADESAENCDK	SevahrfkdlgeenfkalvliafaqyLQQCPfedhvklvnevtefa|k|tcvaDE	5–57 (51)	9.4	9.4	9.1	−9.0	304.9	76.5	5	α-α-corner
			CPfedhvklvnevtefa|k|tcvaDESAENCDKSlhtlfgdklctvaTL	34–80 (51)	10.9	10.9	32.5	−28.7	171.0	53.0	10	α-α-corner

PTM SEQQ—protein fragment with identified PTM; Locus coordinates of the beginning and end of the helical pair containing the amino acid with PTM. In parentheses, the coordinate is modified amino acids; Np—the length of the waist between two helixes; d—interplanar distance, r—minimum distance between the axes of the helixes; θ—torsion and ϕ—planar angles between the axes of the helixes; S—area, P—perimeter of the polygon of intersection of helix projections; Motif-type helical pair.

**Table 3 molecules-25-03144-t003:** The biological role of target proteins and the possible role of PTM.

Protein Name	PTM Position*	Binding Partner	Binding Site Position, a.a.	Structural Localization of PTM and Binding Site	The Role of Partner in Oncogenesis	Estimated Role of PTM
Vitamin d-binding protein (VTDB)	K370-ac K114-ac	Vitamin d metabolites	35–49	Spatially removed	Epidemiological studies suggest that the risk of developing colorectal cancer (CRC) is associated with a decrease in the level of 25-hydroxyvitamin d (25 (OH) d) in the blood [21,22]. However, little is known regarding the role of VDBP in colorectal carcinogenesis [21].	Acetylation of VTDB is a response to the processes accompanying oncogenesis, including an increase in the level of 25 (OH) d, actin, C5a
		Actin	373–403	Spatially brought together	In the case of CRC, cells undergo changes in the cytoskeleton formed by actin, and cell adhesion decreases during invasion [23].	
		C5a/C5a des Arg	130–149	Spatially removed	The complement factor C5a is involved in the processes of tumor formation and profiling of cells [24].	
Complement C4-A (CO4A)	K370-ac	Immune aggregates or protein antigens	1125	Spatially removed	CO4A is involved in the formation of immune complexes.	Regulation of binding of immune aggregates or protein antigens, the content of which in the blood increases in response to tumor growth.
X-ray repair cross-complementing protein 6 (XRCC6)	S77-p	XRCC5 and DNA	31	Spatially brought together	XRCC5/6 dimer is involved in DNA repair mechanisms. DNA modification and binding sites in the protein structure are in the same domain and in close proximity to each other (UniProtKB - P12956 (XRCC6_HUMAN) https://www.uniprot.org/uniprot/P12956)	One of the main etiologies of the onset and development of cancer are genetic polymorphisms, which are associated with the regulation of cell proliferation and differentiation. Cells are likely to react to genetic damage and their ability to maintain genomic stability using the DNA repair mechanism through the XRCC5/XRCC6 dimer [25] is of paramount importance.
Plasma protease C1 inhibitor (IC1)	Y297-p	Chymotrypsin	465–467	Spatially removed	IC1 is involved in the regulation of the activity of the C1r or C1s complex and can play an important role in the activation of the complement system. Very efficient inhibitor of FXIIa. Inhibits chymotrypsin and kallikrein	Tyrosine phosphorylation of IC1 is probably the body’s response to the processes that accompany oncogenesis.
Serum albumin (ALBU)	K223-ac K341-ac K286-ac K75-ac Y287-S82-p K257-glygly	Most anions, hormones, heme, and lipophilic xenobiotics [26]	218–257 (Site I)	Spatially brought together	The albumin/ligand complexation plays an important role in transport, regulation of metabolite and xenobiotic activity. Albumin binds most anions, independent of the hydrophobic character of the ligand side group.	Modification of lysine residues in ALBU may be caused by their increased esterase activity in patients with CRC. Enhanced non-specific phosphorylation of ALBU, especially in warfarin-binding site, is an effect of the impaired cell signaling network significantly contributing to tumor growth and development.

* PTM Position indicates locus amino acid with PTM in the protein sequence (ID PDB) (Appendix A).

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
