# Peer review of "Super Secondary Structures of Proteins with Post-Translational Modifications in Colon Cancer"

_molecules, 2020, doi:10.3390/molecules25143144_

Round 1
Reviewer 1 Report
Tikhonov et al described the analysis and properties of PTM signatures through analysis of human CRC blood samples and other proteins using a computational approach. I think this is an interesting topic in the field of PTM research. Nevertheless, in the original manuscript, the author tried to claim something, but it was difficult to grasp clearly. Human CRC blood sample analysis and previous research results are complicatedly mixed, making readability much worse. For readers' readability, I think it should be summarized or srted as (1) proof of approach, (2) analysis of results, and (3) comparison using different models. In addition, the author should convey a clearer intent in the abstract or conclusion section. In manuscript, the author tried to provide a lot of information overall, and there is no doubt about the results analysis, but it is difficult to recommend publishing a paper in the current manuscript version.
In order to improve the manuscript in the future, I suggest the following.
[1] Even if the PTM or research related contents written in Introduction or result and discussion are general descriptions of PTM, the reference should be clearly marked for the reader.
For example, the following sentences do not contain any reference.
- Line 36-38: Modern advances in systems biology and proteomics have increased interest in studying the role of post-translational modification (PTM) of proteins in pathology development mechanisms.
- Line 38-40: The role in the implementation of biological activity of the protein has been found to be due to the three-dimensional (3D) structure and folding features.
-Line 42-43: PTMs of proteins affect enzymatic activity, protein localization, protein-protein interactions, regulation of signalling cascades, DNA repair, and cell division
-Line 45-48: Methods for detecting PTM can be divided into two groups, the first of which is unambiguous, identifies a specific type of PTM, and is aimed at detecting a wide range of PTMs (10 or more types) in proteins.
-Line 48-49: This group is represented by immune, radio, and fluorescence detection 48 methods using affinity probes.
These statements are common, and the author can clearly understand them, but for the reader, it is necessary to add a reference to the results of previous studies. Likewise in the later sentences for Introduction and discussion, the author should clearly mark the reference if previous research was cited.
[2] Line 52 and line98: Isn't the abbreviation for "Multidimensional Protein Identification Technology" "MudPIT"? And the full name for the abbreviation should appear first on line 52, not line 98.
[3] Line 54-57: “In this study, we used the ultra-high-resolution HPLC-MS/MS approach followed by bioinformatics analysis to identify and characterize PTM loci in protein structures associated with the development of colorectal cancer (CRC).” It is advisable to enter this sentence at the end of the introduction for context flow and reader readability.
[4] Line 105-106: “Structural motifs of proteins containing PTM in blood samples of patients with colorectal cancer” What does this sentence mean?
[5] Line 146 (figure caption): The perimeter (P) is not shown in the figure.
[6] Line 166 (in box): “Search for sections of spirals of the following types: α-spiral, 310-spiral, and π-spiral.”
- In the sentence, 10 in "310-spiral" requires a subscript.
- “the projections of the spirals and the axes of the spirals do not intersect (S ≈ 0 Å2, P ≈ 0 Å); It should be written in superscript in the area unit (S).
[7] Authors need a consistent expression of words. For example
- “alpha-helices” and “α-helices” are used interchangeably.
- “aa-corners”, “αα-corners”, and “a-a corner” are used interchangeably.
- In the amino acid number notation, "aa 373-403 (line 342)" and "379-402 a.a. (line 345)" are used interchangeably.
[8] Lines 317 and 324: The author should clearly indicate for “a.o.”
[9] In Table 2: The protein name, not the accession number, was entered in UniProt AC.
[10] Line 412: ”Supplement 2” should be marked “Supplement materials 2”.
[11] Line 491 and 496: 300 A pore size, 60 A pore size… What unit is A of pore size?
[12] Line 531-532: "In the present study, proteins containing tryptic peptides of a certain type of modification were selected from the PDB [40]." The author should clarify what the protein structure is.
[13] Line537-540: “As a representation of the designation of 537 secondary structures: H - α-helix; B - residue in isolated β-bridge; E - extended strand, participates in β ladder; G - helix 310; I π-helix; T is hydrogen bonded turn; S is bend; and C - coil.”
What does this point to? It seems to be more suitable for figure caption. In addition, the 10 in “G-helix 310” should be subscripted.
[14] I think it is not appropriate to enter the website address in Reference 21 (https://www.uniprot.org/uniprot/P12956) and reference 25 (https://www.uniprot.org/uniprot/P05155). It should be changed to previous literature.
Author Response
Уважаемый редактор,
ответ для рецензента был прикреплен.
С уважением,
Авторы

Reviewer 2 Report
The manuscript “Super secondary structures of proteins with post-translational modifications in 2 colon cancer” submitted by Dmitry Tikhonov and colleagues reports a comparative HPLC-MS/MS study of blood samples from patients with colorectal cancer in which the authors identified several PTM in a set of proteins. Based on their findings, the authors claim that these modifications are present in specific structural motifs (termed by the authors as helical pairs). They also claim that these helical pairs are represented on solvent and characterized by a large area of solvent. Overall, the data provided in the manuscript is potentially interesting for the field. Therefore, this reviewer supports publication after a revision to address the following major and minor concerns.
1) My main concern is regarding the limited number of PTM considered in the present work. The authors based their conclusions on a reduced number of identified PTMs (11 modifications) found in only 6 different proteins that were identified experimentally by the authors using HPLC-MS/MS. In my opinion, the authors should provide a more extensive analysis, perhaps, based on the analysis of databases to support the conclusion that such PTMS occur mainly at these motifs.
2) A second major concern is that the authors considered only three different types of PTM (Lys Acetylation, S/T Phosphorylation, and Lys GlyGly) in their study. However, other different modifications have been identified previously in proteins derived from cancer patients (such as oxidation, methylation or hydroxylation,…). Are such modifications also present in helical pair structures? This should be explained and discussed in the manuscript. Otherwise, the title of the manuscript need to be more specific for the particular type of modifications addressed in the present study.
3) Lines 166-167. This text can be moved to “Materials and Methods” section or provided as supplementary material.
4) I would recommend the authors to rewrite some of the sections of the manuscript. For example; Section 2.4 is somehow too extense and can be shorted and summarized.
5) Figures 1 and 2 can be combined into a single figure.
Author Response
Dear Editor,
a response for the reviewer was attached.
Kind regards,
Authors

Round 2
Reviewer 1 Report
Based on the modified manuscript and response from the author I am satisfied that the work is suitable for publication
Reviewer 2 Report
My recommendation is to reject the manuscript. After revision, it appears that publication would be premature at this time. The number of PTM and experimental data included in the study is not sufficient to support conclusions as currently presented.